# Climate Change and the Rise of Emerging and Re-Emerging Infectious Diseases in Africa: A Literature Review

**DOI:** 10.3390/ijerph22060903

**Published:** 2025-06-06

**Authors:** Robert Kwame D. Agyarko, Dennis Kithinji, Kizito Bishikwabo Nsarhaza

**Affiliations:** 1The African Risk Capacity, Johannesburg 2157, South Africa; 2World Health Organization Regional Office for Africa, Brazzaville P.O. Box 06, Congo; nsarhazak@who.int; 3Medright Consulting Ltd., Maua 60600, Kenya; 4Department of Medical Laboratory Sciences, Meru University of Science and Technology, Meru 60200, Kenya

**Keywords:** climate change, environment, health hazards, infectious diseases, zoonosis, emerging diseases, re-emerging diseases, one health, Africa

## Abstract

Climate change is among the most significant challenges of the 21st century, with global warming, heat stress, floods, and drought occurring in various parts of the globe, including Africa. The impact of climate change on health is becoming increasingly substantial on the African continent due to weaknesses in economies, infrastructure, and healthcare systems. This review explores the relationship between climate change and emerging and re-emerging infectious diseases in Africa and highlights possible solutions. It was conducted by summarizing and synthesizing related information from relevant scientific publications. This review reveals that climate change significantly contributes to Africa’s emerging and re-emerging infectious diseases, including Ebola, Marburg, Lassa fever, dengue fever, malaria, and cholera. The impact of climate change on infectious diseases is variable across the continent, with countries in West and Central Africa experiencing more significant climate change-instigated disease burdens. Multisectoral cooperation between climate change actors, environmental health researchers, policy makers, and political leaders centered in the One Health approach is necessary to develop and implement resilient interventions for climate change-induced emerging and re-emerging infections and related health hazards.

## 1. Introduction

The change in global climate systems owing to excessive greenhouse gas emissions is among the greatest challenges of the 21st century [1], and has significant implications for human health. Climate change impacts developing countries the most [2]. It transcends boundaries to contribute to extreme weather-induced disasters, posing systemic risks to economies, water and food systems, agriculture, infrastructure, and public health [3]. Additionally, it can aggravate over half of human pathogenic diseases through its effects on transmission pathways, resulting in pathogenic diseases, mainly bacterial and viral infectious diseases [4]. Therefore, climate change and the emergence and re-emergence of infectious diseases are interconnected.

Africa contributes the least to greenhouse gas emissions, yet it is one of the most severely climate change-impacted continents [5]. Climate change exacerbates health challenges such as malnutrition, water scarcity, and inadequate sanitation, which may contribute to the emergence and re-emergence of infectious diseases such as Zika, Rift Valley Fever, cholera, and Ebola and may compound existing ones like malaria [6,7,8]. Therefore, the health implications of climate change in Africa are significantly concerning.

Climate change significantly affects infectious diseases endemic in Africa such as Ebola, malaria, dengue fever, and cholera due to Africa’s diverse ecosystem [8]. Rising temperatures and changing rainfall patterns create favorable conditions for disease vectors such as mosquitoes and ticks [7,8,9,10]. Some diseases spread to new regions due to changes in temperature and rainfall patterns [8,11,12]. Consequently, vector-borne diseases are emerging at a high rate, with nearly a quarter of them comprising zoonotic diseases that can be traced to wildlife origins [13].

The Intergovernmental Panel on Climate Change (IPCC) noted that Africa is highly vulnerable to the health effects of climate change as early as 2001 [14]. It attributed this vulnerability to six factors, namely limited water resources; food insecurity from declining agricultural production; threats to natural resource productivity and biodiversity; vector- and water-borne diseases driven by inadequate health infrastructure; coastal areas vulnerable to sea-level rise; and worsening desertification [14]. However, previous studies have mainly focused on the impacts of climate change on health in the global North [15,16]. Therefore, similar studies are needed in Africa [17]. This narrative review summarizes and synthesizes the available scientific evidence on the relationship between climate change and emerging and re-emerging infectious diseases in Africa to contribute to a better understanding of this complex relationship, reveal evidence gaps, and possibly inform the development of policies for the prevention of infectious diseases through climate-based interventions.

## 2. Climate Change and the Rise of Emerging and Re-Emerging Infectious Diseases in Africa

### 2.1. Making the Case: The Impacts of Climate Change and Climate Hazards on the Disease Burden in Africa

United Nations Climate Change [18] observes that the warmer temperatures and higher rainfall increase in Africa facilitate the emergence of new diseases in regions where they were previously not present. The increased incidences of vector-borne diseases such as malaria and dengue are among the most significant impacts of climate change on human health in Africa since 3% of malaria-related deaths and 4% of dengue-related deaths are attributable to climate change [17]. Mosquitoes, which are the main vectors of these diseases, thrive best in environments with high temperatures and increased rainfall, which are their ideal breeding conditions [19]. The continued warming and decreasing rainfall over northern and southern Africa and the increased rainfall over the Sahel region provide such ideal breeding conditions [20]. Since projections foresee a rise of more than 2 °C in the mean annual temperature in Africa [17], the burden of vector-borne diseases is likely to increase.

Climate change is also impacting the availability of clean water and sanitation, which potentially increases the risk of water-borne diseases such as cholera, dysentery, and typhoid fever [21,22]. Scarcity of water resources due to prolonged droughts, such as in East Africa, or floods, such as in West Africa, leads to inadequate water for drinking, sanitation, and hygiene, increasing the incidence of diarrheal diseases [23]. Africa has the largest burden of food-borne diseases. This is mainly driven by the unavailability of safe water, which is worsened by global warming [23]. The limited climate resilience across Africa worsens access to clean water, hygiene infrastructure, and sanitation [24], thus increasing the incidence of water-borne diseases.

#### 2.1.1. Climate Change and Re-Emerging Infectious Diseases

Climate change is influencing the emergence and re-emergence of infectious diseases worldwide, including in Africa [13,16,25,26]. Specifically, climate change is impacting the geographic distribution and abundance of vectors like mosquitoes and ticks, and animal hosts such as bats and rodents [25,26]. Besides spreading infections, the vectors and hosts affect human behavior and overall health. Therefore, the re-emerging infectious diseases increase the pressure on the few resources available for disease control and prevention in Africa. The lack of public health resources compromises the comprehensiveness of public health responses to emerging and re-emerging diseases [27], leading to prolonged presence and further spread of the infections.

Climate change substantially influences the occurrence of zoonotic diseases in humans. For instance, the Ebola, Marburg, and Lassa viruses originate from bats and rodents and spread to humans through contact with infected bodily fluids [28]. Climate change alters the ranges and behaviors of these animals, making it easier for them to come in contact with humans and increasing the likelihood of disease transmission [29]. Humans also invade forests and rangelands through deforestation and changes in land use in search of natural resources due to climate change-instigated deficiencies in the resources in human habitats [30]. Therefore, as climate change progresses it predisposes Africans to higher risks of zoonotic diseases.

Increased contact with non-human primates is contributing to the re-emergence of zoonoses such as yellow fever [31], which is transmitted to humans by mosquitoes after infecting non-human primates. Temperature and rainfall patterns affect mosquito abundance and distribution, while deforestation creates new habitats for non-human primates, thus enhancing the spread of viral infections like yellow fever [30]. Therefore, increasing changes in the activities of humans and non-human primates in response to environmental changes present threats for the emergence and re-emergence of zoonotic diseases.

#### 2.1.2. Climate Hazards and Specific Infectious Diseases in Africa

The correlation between climatic hazards and infectious diseases in Africa is complex and varies by region due to diverse climatic conditions, environmental factors, and socioeconomic determinants [18]. The distribution of the climate change-influenced emergence of Ebola, Marburg, and Lassa fever in Africa varies across the continent [32,33]. Therefore, the incidences of hemorrhagic fevers due to emerging and re-emerging viruses in various regions in Africa are as diverse as the climatic conditions of the regions.

Climatic factors play a role in Ebola outbreaks in Africa even though the Ebola virus is mainly transmitted through contact with the bodily fluids of infected persons [34]. Changes in rainfall, vegetation, and forest cover affect the frequency and severity of Ebola outbreaks. For example, heavy rains cause an abundance of fruits, which provides food for natural hosts of the virus such as bats (Figure 1) [35]. During these periods, heightened interactions between humans and infected animals can lead to outbreaks [31]. Moreover, deforestation and encroachment into forested areas can contribute to the emergence and spread of Ebola [30,36]. Therefore, climate change and the resultant human activities increase the risk of Ebola in Africa.

The Marburg virus, which is in the Filoviridae family with Ebola, causes outbreaks that show some correlations with climatic factors. Marburg outbreaks are more common in locations with many natural hosts, such as fruit bats [37]. Climate-related factors such as changes in rainfall and temperature can trigger ecological changes, including alterations in the distribution and behavior of bats [38]. For example, a species of bat, *Rousettus aegyptiacus*, increases its activity in residential areas during cold seasons, thus increasing the likelihood of transfer of the Marburg virus from them to humans [39]. Outbreaks have been reported in different countries, each with unique characteristics, including climatic and ecological factors, making it challenging to identify consistent regional patterns.

The presence of the Lassa virus, a member of the family Arenaviridae that is rodent-transmitted, mainly in West Africa as a cause of hemorrhagic fevers in humans, could be influenced by climate change [40]. While the relationship between climatic hazards and Lassa fever is not as well established as with Ebola and Marburg, correlations between climate change and the incidence of Lassa fever have been observed [41]. During the dry season in West Africa, which is associated with an increase in Lassa fever cases, rodents—the primary carriers of the virus—may seek food and shelter in human dwellings due to a scarcity of food resources, increasing the chances of human contact and transmission [42].

Meteorological factors influence the incidence of Mpox, an infectious zoonotic disease caused by the Mpox virus of the family Poxviridae. Key meteorological variables such as temperature, humidity, and precipitation significantly affect the transmission dynamics of Mpox [43]. Warmer temperatures and changing rainfall patterns may alter the distribution and survival of the primary reservoir hosts—rodents—potentially increasing human exposure. Additionally, deforestation and habitat loss, which are exacerbated by climate change, drive wildlife, including rodents and non-human primates, closer to human settlements, increasing the risk of zoonotic spillovers such as Mpox [31]. A recent study that analyzed data from 52 countries on daily confirmed Mpox cases while considering climate types, the mean surface air temperature, and precipitation found that temperature and precipitation significantly impacted Mpox prevalence in South America [44]. Hence, Mpox outbreaks are likely to increase as climate change causes adverse weather conditions.

Climate hazards significantly compound malaria in the malarious regions of Africa, where malaria is already a leading cause of morbidity and mortality. Most of the two million additional malaria cases in 84 malaria-endemic countries between 2020 and 2021 were in Africa, with countries in Western Africa leading in age-standardized incidence rates and mortality rates [45]. Malaria epidemics often happen after periods of heavy rainfall and rising temperatures in the tropics, which allow malaria-carrying mosquitoes to survive at higher altitudes [18]. Malaria incidence rates in Africa largely declined from 2000 to 2018 in a temperature-dependent manner; countries such as Uganda and Mozambique reported high incidence rates [46]. Climate-based modeling predictions show that the prevalence of malaria is likely to diminish in the near and far futures in West Africa as the climate in the Sahel region becomes unsuitable for *Anopheles* mosquitoes [47]. Therefore, the fight against malaria in Africa is likely to benefit from climate change.

Dengue fever, another mosquito-borne viral disease, is also becoming a significant public health threat in Africa in the context of climate change. Temperature and dengue incidence are positively correlated [48]. Hence, the prevalence of dengue is likely to increase as global warming occurs. According to the World Health Organization (WHO) [49], dengue incidences have grown dramatically worldwide in recent decades, with cases reported to the WHO increasing from 505,430 in 2000 to 5.2 million in 2019. Recent dengue fever outbreaks in East and West Africa have been linked to rising temperatures and rainfall [50]. Critically, the risks of dengue fever may change in tropical and subtropical areas as vectors thrive in environments that are altered by climate change [48]. Therefore, dengue-endemic regions are expected to expand in the face of climate change, which favors the vector dynamics of *Aedes* mosquitoes.

Cholera, a water-borne disease, is also closely linked to climatic hazards. Over a quarter million cases of cholera were reported in 19 African countries between January 2023 and January 2024 [51]. Frequent and severe floods contaminate water sources with fecal matter, increasing the risk of cholera outbreaks [52]. In displacement camps established to respond to climate change-related disasters, cholera is the most common cause of outbreaks with fecal–oral transmission [53]. Access to clean water and healthcare services is reduced with the destruction of infrastructure due to floods, landslides, and other climate change-related emergencies [52], further exacerbating the spread of cholera. The Southern African Development Community (SADC) recommended the inclusion of climate change-targeting interventions in the multisectoral response to cholera re-emergence as peak numbers of cholera cases and deaths were reported in their member states [51].

### 2.2. Health Systems and Social Implications of Climate Change-Triggered Emerging and Re-Emerging Infectious Diseases

Climatic hazards can lead to disruptions of routine healthcare services and disorient social systems [54,55]. Extreme weather events can displace populations, destroy health facilities, and disrupt healthcare services. Thus, they can decrease the availability of essential health services, with long-term implications for the health of populations [56], especially for vulnerable groups such as children and pregnant women.

The increased burden of infectious disease outbreaks due to climate change can lead to increased healthcare costs, reduced productivity, and decreased economic growth [57]. Agriculture, which is the economic backbone of the West African countries affected by Ebola, significantly deteriorates as farmers abandon their farms to avoid infection [58]. According to Smith et al. [57], the Ebola epidemic in West Africa demonstrated the serious and unanticipated economic toll of an emerging infectious disease. For example, Liberia’s Gross Domestic Product (GDP) decreased from 8.7% to 0.7% due to Ebola and lowered commodity prices while in Sierra Leone (excluding iron ore) the GDP decreased from 5.3% to 0.8% [57]. Therefore, climate change-related Ebola outbreaks are economically costly and can exacerbate existing social inequalities, leading to a cycle of poverty and ill health.

### 2.3. Key Challenges and Trends in Climate Change-Triggered Infectious Diseases

Reports of flood events have increased, with Nigeria experiencing its worst flood in 2022, losing more than 610 people [59]. In March 2023, severe flooding killed dozens and affected more than 300, 000 in Ethiopia and Somalia [60]. Such disaster events lead to loss of life, destruction of property, displacement of people, and increases in infectious disease burdens, as experienced by countries like Guinea, Mali, Sierra Leone, Burkina Faso, Ghana, Niger, the Central African Republic, and Nigeria [59]. Hence, heavy floods not only increase the emergence and re-emergence of infectious diseases but also limit countries’ capacities to respond to the diseases due to the economic implications of flooding.

Perpetual droughts co-occur with floods in Africa, which further threatens humanity, fauna, and flora. Consequently, the prevalence of vectors that thrive in dry conditions increases. For example, sandflies breed in dry conditions and bite humans to transmit Leishmania species, causing Leishmaniasis [61]. Rodents migrate to homesteads during drought seasons in search of water and food, thus increasing the occurrence of infections such as hantavirus, the plague, and leptospirosis [62]. Fecal–oral transmission of waterborne pathogens such as *Vibrio cholerae* and *Shigella* species increases during droughts due to the scarcity of drinking water. Thus, diarrheal diseases are common during droughts [63]. As a result, the burden of emerging and re-emerging diseases is increasing as climate change-driven drought seasons hit Africa, considering that most countries have poor sewerage and sanitation systems.

## 3. Mitigating the Impacts of Climate-Induced Health Hazards

African countries must undertake deliberate actions to mitigate the impacts of climate-induced health hazards. A One Health approach that transcends human health, animal health, and the environment is recommended for coordinated mitigation and adaption [64]. The first action would be implementing climate-resilient health systems [65]. African states must integrate climate adaptation into health system planning, resource allocation, and service delivery, including investment in infrastructure and health workforce development, ensuring access to essential medicines and supplies and establishing effective disease surveillance and response systems [66]. Secondly, community-based health programs should be strengthened since they provide a country’s first line of defense against infectious diseases. Governments should prioritize inclusive community-based health programs that engage local communities in identifying, preventing, and managing climate-induced health hazards [67].

Thirdly, fighting climate change-triggered infectious diseases necessitates improving water, sanitation, and hygiene (WASH) infrastructure. African countries should invest in climate-resilient WASH infrastructure, including sustainable safe water supplies, sanitation facilities, and hygiene promotion programs, to prevent water-borne diseases and improve health outcomes [68]. Protecting the WASH infrastructure from climate hazards calls for the establishment of early warning systems that integrate meteorological, environmental, and health data to enhance the detection of climate-induced health hazards and to respond promptly and effectively [69]. Therefore, early warning systems slow the spread of infectious diseases by protecting and optimizing the utility of the available resources.

Additionally, African countries should promote research and innovation in developing new vaccines, diagnostic tools, and treatments for climate-sensitive diseases and should explore innovative approaches to health service delivery [70]. They can leverage regional and international collaboration to learn from countries that affordably established diagnostic and vaccine manufacturing systems in response to the COVID-19 pandemic for the cost-effective manufacture of vaccines, tests, and drugs [71]. This includes establishing partnerships with research institutions, donor agencies, and international organizations to develop and implement climate-resilient health programs [72]. Developments should consider social determinants of health such as poverty to ensure accessibility and affordability of the innovative drugs, vaccines, and tests for most Africans in the African Continental Free Trade Area (AfCFTA), thus promoting universal healthcare and reducing dependency on imports. Overall, all interventions require sustained political commitment, investment, and collaboration at both the national and international levels for long-term results [73]. Therefore, multisectoral stakeholders should collaborate with African governments to design systems that will address the effects of climate change and create robust systems to withstand those effects and possibly reverse them (Figure 2).

## 4. Conclusions

This review highlights the relationship between climate change and emerging and re-emerging infectious diseases in Africa. It shows that climate change and climate hazards have substantial effects on the emergence and re-emergence of infectious diseases, particularly vector-borne diseases and diseases with fecal–oral transmission, in Africa. While some regional similarities exist in the correlations between climatic hazards and infectious diseases in Africa, the impact can vary by region due to the diversity of environmental, economic, and political factors. Climate hazards disrupt routine healthcare services, economic systems, and social structures. Thus, a multisectoral approach that involves strengthening public health systems, improving access to clean water and sanitation, and developing early warning systems to identify and respond to disease outbreaks is essential. Strengthening regional and international collaboration related to research, policy development, and technological advancements and political commitments related to establishing resilience to climate-induced health hazards in Africa should also be prioritized.

## Figures and Tables

**Figure 1 ijerph-22-00903-f001:**
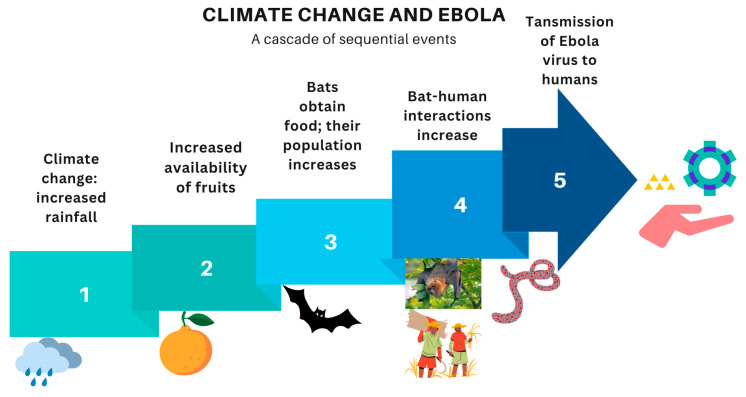
An illustration of how heavy rainfall due to climate change causes Ebola outbreaks in humans.

**Figure 2 ijerph-22-00903-f002:**
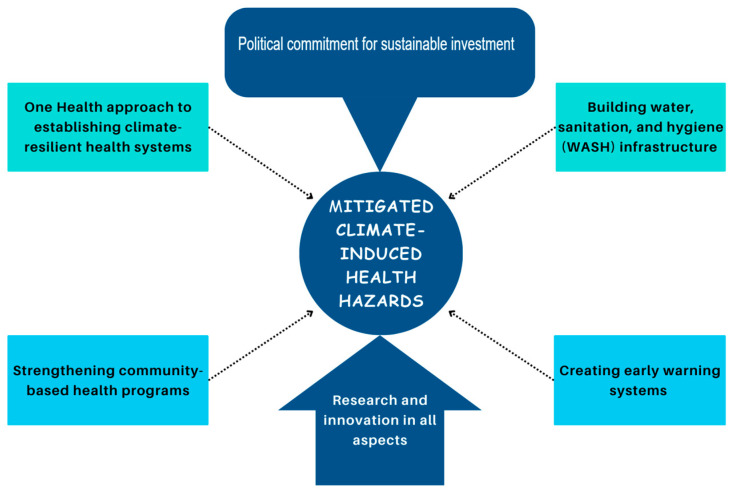
A mind-map showing a multi-pronged approach to mitigating climate-induced health hazards.

## Data Availability

No new data were created or analyzed in this study. Data sharing is not applicable to this article.

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
