# Peer review of "Climate Change and the Rise of Emerging and Re-Emerging Infectious Diseases in Africa: A Literature Review"

_ijerph, 2025, doi:10.3390/ijerph22060903_

Round 1
Reviewer 1 Report
Comments and Suggestions for Authors
Reviewer’s comments for ijerph-3581047
Title
Climate change and the rise of emerging and re-emerging infectious
Diseases in Africa, a literature review
Overall comments
The authors touched the important topic and reviewed a good number of related updated literatures. The manuscript was divided into sub-headings with specific topics in serials. Although there are repetitions in some descriptions, the manuscript was composed nicely as a literature review and brought a compact information. The references may need some revisions according to the Journal Format. The Manuscript is acceptable after minor revision.
For corrections and suggestions
Abstracts
- There is overlapping of sentences and meanings such as “relationship between climate change and emerging/re-emerging infectious diseases in Africa” “possible solutions” “suggestions that can ameliorate the problem”: In Line 13-14;” This review explores… highlights the possible solutions” (I guessed it is for objective) and In Line 15- 18; “It was conducted by… ameliorate the problem (It is for methodology)”.
- I suggested to rewrite these 2 sentences to make clear and compact abstract.
- In Key words, the authors included “health hazards”. Although this word was mentioned in the text of the manuscript, it should also be mentioned in the abstract. or it can be deleted.
Introduction
- I suggested to include some description/examples for emerging and reemerging infectious diseases prevailing in Africa
- In Line 49-50, it was described the reference as “7,10”. I found there is no Ref. 9. Please confirm whether the authors would like to mention the reference as “7-10”.
- Climate Change and rise of …….
- In 2.1 Disease Burden, It will be stronger evidence for the rise of infectious diseases if the authors add the disease burden figure (eg. incidence rates/trends/numbers of climate change-related diseases)
- In 2.2 Climate change and Re-emerging ID”
- I suggested to mention “emerging” in the subheading as “emerging and reemerging infectious diseases as the authors mentioned both areas in the text.
- In third paragraph page 4, please make uniformity for “MPOX” and Mpox. Mpox is preferrable term.
- In 2.3 Key Challenge , please correct “Vibrio cholerae” not Vibrio cholera.
- 4 Mitigating the Impact of Climate-induced health hazards
I suggested to divide this session separately as 3. not under” 2. Climate change and rise of ---” as it summarizes and synthesizes the suggestions and possible solutions mentioned in the literatures. The authors described in the abstracts that this review explores 2 areas; “the relationship between climate change and infectious diseases” and “Solutions and suggestions that can ameliorate the problem”. By making different session as “3” before conclusion will make the manuscript for clearer and serial description to fulfilling the main objectives of this review.
- Conclusion- Fine
References-
- Pls check in accordance with the journal format.
Reviewer 2 Report
Comments and Suggestions for Authors
- Keywords: include One Health, Zoonoses
- Introduction: Line 37-39: sentence appears hanging; line 61: why restrict similar studies to Africa and not to other continents in the South?
- Climate change and rise of emerging and re-emerging infectious diseases in Africa. Line 73-75: rephrase sentence; Lines 176, 228, 232: italicize Anopheles, Aedes, Shigella, etc.; line 228: either sandfly breeds or sandflies breed.
- General comments: i. There appears to be repetitions of phrases, ii. reference to winter may not be applicable to the tropical and sub-tropical regions of Africa, iii. there is need to include a brief on emerging and re-emerging diseases in general to provide a context for the impact of climate change, iv. the impact of climate change and emerging diseases on SDGs should be highlighted, v. there is need to capture the poor emphasis on environmental issues in Africa as well as poor research infrastructure.
Generally good
Reviewer 3 Report
Comments and Suggestions for Authors
IJERPH review
Climate change and the rise of emerging and re-emerging infectious diseases in Africa
This is an important topic. The authors did not mention some additional critical aspects. In section 2.2, they briefly address access to health care, but this should be addressed more thoroughly. Additionally, infections that could be addressed better with vaccination should be discussed.
Another topic that should be discussed more thoroughly especially since this is about Africa would be heat stress and folate and vitamin B deficiency on immunity. Hematopoietic effects and immunity should be mentioned along with diet or elsewhere.
Climate issues also are causing an increase in psychological stress which will interfere with immunity to pathogens this is not mentioned at all.
A table might be useful to connect particular pathogens with certain aspects connected to climate issues like the water contamination, poor diet, etc. Some of these connections are mentioned in the text but a table connecting them would help readers to focus on the connections.
